# Influence of Pozzolans and Hemp Shives on the Properties of Non-Autoclaved Foamed Concrete

**DOI:** 10.3390/ma16020591

**Published:** 2023-01-07

**Authors:** Abdelrahman Mohamad, Fouzia Khadraoui, Mohamed Boutouil, Daniel Chateigner

**Affiliations:** 1COMUE NU, Laboratoire de Recherche, Builders Ecole d’ingénieurs, 1 Rue Pierre et Marie Curie, 14610 Epron, France; 2CRISMAT, CNRS UMR 6508, ENSICAEN, IUT Caen, Université de Caen Normandie, Normandie Université, 6 Bd Maréchal Juin, CEDEX 4, 14050 Caen, France

**Keywords:** foamed concrete, hemp shiv, mechanical properties, thermal conductivity, environmental impact

## Abstract

In this study, new foam concretes incorporating hemp shives without the use of autoclaving have been developed and studied. Several protocols and parameters were investigated. Firstly, the influence of the addition of pozzolanic additives on the resulting density, mechanical behaviour and thermal conductivity was examined. Secondly, the effects of the incorporation of hemp shives at 5, 10 and 15 vol% on the previous three properties in such concretes were investigated. Moreover, economic cost and CO_2_ emissions were estimated to outline an optimized formulation of non-autoclaved biobased foam concretes. First, the target characteristics in terms of compressive strength (minimum of 2 MPa), thermal conductivity (less than 0.2 Wm^−1^K^−1^) and density (800 kg/m^3^) were achieved. It was noted that pozzolanic additions slightly improved the mechanical and thermal strength of non-autoclaved foamed concrete, while the addition of hemp shives improved the thermal strength but had an unfavourable effect on the mechanical strength. Moreover, both reduced the CO_2_ emissions.

## 1. Introduction

In recent years, lightweight, aerated and foam concretes have been used in construction for their manyfold benefits, such as concrete for thermal insulation, renovation, screeds, etc. Generally, the Foamed Concrete (FC) is based on the incorporation of air or gas bubbles in the mineral matrix and is formulated to reach a low density ranging from 400 to 1800 kg/m^3^ and a low compressive strength ranging from 1 to 10 MPa. Therefore, it is a non-structural material [1] with a very low thermal conductivity (between 0.09 and 1.2 Wm^−1^K^−1^) compared to usual concretes (typically from 1.6 to 2.2 W m^−1^K^−1^). FC plays an important role in building thermal insulation [2]. Additionally, FC provides more advantages, such as acoustic insulation, fire resistance, lowering construction cost and producing low self-weight and eco-friendly materials. Indeed, more than half of its volume is made up of air, which reduces the amount of raw materials used [3,4,5]. Usually, the production of foamed concrete is achieved by mechanical means. There is the preformed method, where a foaming agent is used to create foam before being added to the mix, and there is the mixed method, where air bubbles are created by adding the foaming agent directly into the cementitious matrix [6]. In general, two types of curing are used to ensure the curing of foamed concrete, autoclaving and non-autoclaving. However, autoclaving at high temperature and high pressure significantly increases production energy consumption [1], with an estimated embodied energy of 340 kWh/m^3^, which corresponds to 168 kg CO_2_/m^3^ of emission. According to Couasnet [7], this embodied energy is near that of conventional concrete, but on the other hand, the elimination of autoclaving ensures that the foam concrete cures in the open air or in a damp room. Therefore, the foamed concrete discussed in this article was not autoclaved and was manufactured using the preformed foam method.

Some of the available experimental developments that consider replacing a portion of the cements are focused on evaluating the use of fly ash, metakaolin and Ground Granulated Blast Furnace Slag (GGBFS), which is known for its pozzolanic activity, as it promotes the hydration process [8,9,10,11,12,13]. The use of fly ash can reduce the amount of cement required per cubic meter by 50% and the hydration temperature by 40% [14] as well as increasing compressive strength at an early age due to reduced bubble sizes [15].

Several studies have focused on the performance and behaviour of FC with different raw material additives, such as pozzolans, phosphogypsum and polypropylene fibres [1,16,17,18,19], but few have studied foam concrete incorporating hemp shiv (HS) [20].

Most of the published studies in the literature have also focused on the thermal and hygrometric qualities of hemp shiv concrete (HSC) due to the high porous structure of hemp shiv (larger than 70%) [21,22]. Regarding mechanical properties, the compressive strength of HSC remains relatively low compared to other conventional construction materials, often below 2 MPa [23]. HSC has a very high-water absorption capacity due to the porous structure of hemp shiv, which can be more than 300% by mass [24].

Hemp shives are largely used in studies of biobased concretes as a vegetable aggregate. The most important characteristics of hemp concrete are their lightweight, prefect moisture buffer capability, acoustic insulation and low thermal conductivity, which all allow reduced building energy consumption and gas emissions [16,17]. Meanwhile, incentives are proposed for bio-based materials so that they can play an increasingly efficient role in the composition of building materials [25,26,27].

This work attempts to create Biobased Foam Concretes (BFC) for non-structural applications (self-supporting), combining foamed and bio-based concrete technologies. The target characteristics are a compressive strength of at least 2 MPa (the minimum compression to achieve a self-supporting material) and thermal conductivity and density smaller than 0.2 Wm^−1^K^−1^ and 800 kg/m^3^, respectively. BFCs containing hemp shiv as a bio-based raw material up to 15 vol% were investigated, with and without pozzolanic additives and as a function of hemp shive contents. Hemp shiv was used as a substitute to decrease the use of cement, thereby decreasing CO_2_ emissions.

## 2. Materials and Methods

### 2.1. Raw Materials

The materials used to produce BFCs were:Portland Cement CEM I 52.5N, in accordance with the EN 197-1 standard. The absolute density was 3100 kg·m^−3^ and the bulk density was 2200 kg·m^−3^.Ground Granulated Blast Furnace Slag, fulfilling EN 206-1, a by-product from iron production with an absolute density of 2900 kg·m^−3^.Metakaolin (MK) with an absolute density of 2600 kg·m^−3^, obtained by kaolinite flash calcination at approximately 700 °C, the presence of MK acted as an activator that neutralised the retarding effect of GGBFS at the early age of resistance development and improved resistance at later ages [13].Plant materials composed of 95% hemp shives and 5% hemp fibres. A characterisation of hemp shiv particles according to the recommendations of the RILEM TC 236 BBM Technical Committee was investigated [28]. Hemp shiv presented an absolute density of 1400 kg·m^−3^, bulk density of 140 kg·m^−3^, thermal conductivity of 0.048 Wm^−1^K^−1^ and hemp shiv particle size distribution ranging between 0.6 and 5 mm. Figure 1 shows the water absorption of hemp particles and shows that the maximum water absorption of hemp particles was approximately 247%.

Pozzolans are well-known for increasing resistance to sulphate attack, decreasing the heat of hydration, increasing durability and reducing energy cost per unit of cement [29], which is why some pozzolanic additives were used in this study (Table 1).

To improve the workability of the mixture, a superplasticiser (SP), MasterGlenium ACE 550 from BASF, based on modified polycarboxylate and phosphonates was introduced into each specimen to reduce the amount of water used while keeping a good fluidity of fresh mix. A commercial protein-based foaming agent (FAG), Betomouss from SIKA, was used to produce a more stable bubble network, and the density of the foam was approximately 70 kg/m^3^. Foaming agent used, and the HS had a negative effect on the degree of hardening of the concrete [30]. Therefore, an accelerator (Acc), MasterSet AC 555 from BASF, was used to promote the initial setting of the foamed concrete, which improved the stability of air bubbles in the cement matrix and rapidly developed the mechanical resistance. Moreover, a demoulding product was used, whose composition did not contain any solvent. It was used to easily demould the foamed concrete after hardening, especially since foamed concrete is sensitive. The characteristics of the accelerator, superplasticiser and the foaming agent are given in Table 2.

### 2.2. Specimens Elaboration

Eight formulations of foam concretes were prepared, each of them composed of three samples (see Table 3). Two formulations were created without hemp shives (C100P0H0 and C70P30H0), as control, based on those found in the literature to produce a structural insulating foam concrete [31]. A preliminary test of the substitution of cement by hemp shives with amounts from 1 to 30 wt% was conducted. It was found that the maximum content to give a stable foamed concrete without deflation was 15 wt%. Therefore, based on the two control formulations, three different amounts of cement (5, 10 and 15 wt%) were substituted with hemp shives at constant total volume to evaluate mechanical and thermal properties, density, production cost and CO_2_ emissions, as in Page [32].

The composition of all foamed concretes studied in this paper (Table 3) were designed for 1 m^3^ and called CxPyHz, where C, P and H stand for cement, pozzolanic additions and hemp shiv, and x, y and z stand for their content in wt%, respectively.

Due to the high-water absorption capacity of HS (247% of hemp shiv mass), hemp shiv particles absorb water, which reduces the water content that reacts with the mineral binder during the production of concrete containing hemp. In other words, hemp particles absorb the water needed to hydrate the binder [23,32]. This phenomenon leads to chalking of the hydraulic binders at the interface between the hemp particles and the binder [33]. To solve this problem, the amount of water was varied according to the volume.

In Table 3, Wh presents the amount of water absorbed by the hemp particles, Wt presents the total amount of water, Wb presents the amount of water to hydrate the binder, and B is the binder, where Wt = Wb + Wh. In addition, regarding Table 3, Wb/B is always the same whatever the amount of binder, and Wt/B are different to keep a similar workability for all BFCs relative to the amount of hemp added and cement removed.

Hemp shives were added to the mixes to reduce cost and CO_2_ emissions, and to determine their effects in foamed concrete. The preformed foam method was used for the specimen elaboration. It was based, firstly, on the production of a light aqueous foam, which was then progressively added into a mineral suspension. On one hand, the foam was generated by mixing the diluted foaming agent (in a 1:30 ratio by volume) with water. On the other hand, the powdered constituents and the hemp shives were first mixed dry in a mortar mixer with a capacity of approximately 20 L. Then, water with superplasticiser and accelerator were added and the mixture was mixed until a homogeneous paste was obtained. Finally, the mineral mix was added to the foam progressively and mixed until the mixture became homogeneous. The obtained foamed concrete was fluid and placed without any vibration in prisms 4 × 4 × 16 cm^3^ for the testing of mechanical properties and 30 × 30 × 7 cm^3^ for the testing of thermal properties, and all protected from drying out. After 48 h of curing at 20 °C, the specimens were removed from the moulds and stored in a damp room (20 °C, RH > 95%).

### 2.3. Test Methods

The fresh and dry density were measured according to NF EN 12350-6 and NF EN 12390-1 standards, respectively. Three 4 × 4 × 16 cm^3^ parallelepiped samples were dried at the temperature of 60 °C until the weight loss stabilised. Density was then calculated from the stabilised weight value and the volume.

After 7 and 28 days, one specimen was broken in half during a flexural test and the compressive strength (Rc) was measured for both halves. A universal press (IGM^®^ 250KN press) with a load velocity of 0.05 kN/s was used to determine the unconfined compressive strength (UCS) and the flexural strength (Rf). These tests were controlled by adopting the NF EN 679 and NF EN 196 standards.

The thermal conductivity was measured by following the EN 12667 standard. The temperatures applied for measurement were: Ttop = 30 °C (temperature of top plate), TBottom = 10 °C (temperature of bottom plate), Tmean = 20 °C, ∆T = 20.

The images obtained allowed a first observation of the BFC structure and interfaces between sample matrices and HS particles.

## 3. Results and Discussion

### 3.1. Foamed Concrete Density

The main factors affecting concrete density are air content, aggregate density, binder and water content [34]. The density of all samples varied between 670 kg/m^3^ (C100P0H0) and 560 kg/m^3^ (C85P0H15) (Figure 2), meaning that they fulfilled the first development target (density less than 800 kg/m^3^).

Density decreased regularly and by approximately 7% as the hemp shiv content reached 5% (Figure 2), since the density of cement is higher than that of HS. Moreover, by comparing C70P30H0 (including pozzolanic additions) and C100P0H0 (without pozzolanic additions) densities, it was found that the latter slightly decreased with the addition of pozzolans. This could be explained by the raw materials’ densities (GGBFS: 2900 kg·m^−3^ and MK: 2600 kg·m^−3^) being lower than the density of cement (3100 kg·m^−3^).

### 3.2. Mechanical Properties

In general, the compressive strength of foamed concrete largely relates to age, raw materials, porosity and dry density [35,36]. In our samples (Figure 3), the compressive strength increased with age in a ratio Rc7/Rc28 = 0.8, i.e., more than for an ordinary concrete in which such a ratio is observed at around 0.65. This effect was mostly due to the setting accelerator used.

Page et al. [34] remarked that the addition of a superplasticiser improves the interface between the pozzolanic binder and the hemp aggregates, thus, raising the rigidity of the material. Moreover, with the addition of pozzolans, the hydration process was delayed and took longer to stabilise over time [29]. For this reason, the increase in the mechanical strength of foamed concretes with pozzolanic raw materials C70-BFCs was larger than those without (Figure 3). Indeed, the compressive strength at 28 days of C70P30H0 was larger than that of C100P0H0, although it was lower at 7 days. In addition (Figure 3), due to the low cohesion between the hemp particles and the cementitious matrix, the standard deviation of the compressive strength of C85P0H15 was larger than the others, which also made controlling the compressive strength more difficult.

The density of the material decreased with the addition of hemp shiv [22]. In the same way, the low cohesion between the cementitious matrix and the hemp particles led to the creation of new gaps and increased air entrapment, which affected the stiffness of the materials [37]. Therefore, there was a large drop in the strength of the C100P0H0 sample after the addition of hemp shiv, which happened to a much lesser extent in C70P30H0 (Figure 3). Optical microscope images (Figure 4) showed the presence of gaps between the cementitious matrix and the hemp shiv particles, leading to the low cohesion. Consequently, the hemp shiv incorporation in FC led to a significant decrease in compressive strength for C70-BFCs ranging from 24% for C65P30H5 to 37% for C55P30H15, and for C100-BFC, ranging from 52% for C95P0H5 to 63% for C85P0H15. Therefore, the material became more fragile, and the FC specimens without HS exhibited a higher strength compared to those with 5%, 10% and 15% HS. The decrease in the compressive strength by the incorporation of raw plant materials in the biobased concretes has been highlighted by other authors [38,39]. Moreover, the reduction in compressive strength can be caused by the decrease in the cement amount in the mix. Indeed, Chamoin et al. [22] also noted that the compressive strength of hemp concrete is impacted by the used binder; it can be enhanced by optimising its dosage.

As for C55P30H15 and C85P0H15, bio-based foam concretes (BFC) with the same amount of hemp shives but different cement contents, presented significant differences in compressive strength. This is also due to the mix of GGBFS and MK, which can improve the cohesion in the mineral matrix [13]. Moreover, in the case of conventional bio-based concretes, it is better to mix the used binder, lime or cement with additives, due to their ability to impact the final number of hydrates, and to increase the adhesion between the hemp shives and the binder [40].

Flexural strength (Figure 5) followed similar behaviours as the compressive strength, with a noticeable decrease upon HS additions for C100-BFCs ranging from 31% for C95P0H5 to 44% for C85P0H15, and for the C70-BFCs samples, from 21% for C65P30H5 to 30% for C55P30H15. In addition, replacing cement with HS decreased the amount of binders in the formulation. This correlates to Williams et al.’s [8] observations, which show an increase in flexural strength upon binder additions, both in perpendicular and parallel directions. The flexural strength drops exhibited by all our samples upon HS addition compare well with Williams et al.’s results.

To better understand the effect of hemp shiv on flexion and on compression, the ratio Rf/Rc can give us an indication. For example, in the case of fibres, this ratio becomes greater than in the case of concrete, considering that the behaviour of fibres in flexion is much more important than in compression, while the behaviour of concrete in compression is more important, so the ratio becomes smaller. The ratio Rf/Rc of the BFC samples ranged between 0.28 and 0.45. This ratio is larger than that of FC found in the literature, with a ratio between 0.2 and 0.4, and for ordinary concretes with a ratio between 0.1 and 0.2 [41]. Comparing ordinary concretes with foam concretes, the addition of HS reduces the compressive strength more than the flexural strength, so the Rf/Rc ratio is increased. Indeed, the hemp shiv performs better in flexion than in compression [42].

### 3.3. Thermal Conductivity

As an overall behaviour (Figure 6), thermal conductivity of our samples did not exhibit significant differences between 7 and 28 days, with or without pozzolans. The two usual and major factors influencing thermal conductivity were material’s density and the thermal conductivity of the constituting materials of the concretes. On one hand, thermal resistance is inversely proportional to the density of the materials [42,43], and on the other hand, the intrinsic conductivity of HS (around 0.04 Wm^−1^K^−1^ [44]) is lower than that of pozzolans, and the latter two are lower than cement [45].

In our case, the thermal conductivity of the BFCs without pozzolans (Figure 6a) was affected by the incorporation of hemp shives, whereas this behaviour was almost absent for BFC samples with pozzolans (Figure 6b). However, in BFCs, not only density and intrinsic thermal properties of the constituents acted on the concrete’s thermal conductivity. For instance, the HS interface’s gap observed using optical microscopy also contributed to the corresponding air entrapment, leading to a further decrease in thermal conductivity. In addition, comparing C100P0H0 and C70P30H0 conductivities, the thermal conductivity of the BFCs was predominantly affected by the pozzolanic additives, whatever the ageing. This tendency was also observed for the conductivity of the C95P0H5 and C65P30H5 samples. Then, comparing the thermal conductivities of C70P30H0 and C65P30H5, whatever the ageing, almost similar values were observed, with 0.1402 and 0.1333 Wm^−1^K^−1^ at 7 days, and 0.1392 and 0.1312 Wm^−1^K^−1^ at 28 days, respectively, while without pozzolans (Figure 6a) the difference in thermal conductivity is clearly visible between the H0 and H5 samples. Consequently, pozzolans partially impeded thermal conductivity decrease resulting from hemp shiv incorporation. This behaviour was observed for larger HS contents of the BFCs with pozzolans for the two ageings. Consequently, the interaction between pozzolans and HS were responsible for the partial thermal conductivity compensations. Since HS incorporation resulted in incohesive interfaces, the addition of pozzolan might better help filling pores [46] to give rise to slightly larger thermal conductivities, as these latter remained lower than 0.15 Wm^−1^K^−1^ from 7 days of ageing.

Pozzolanic additives in foamed concrete decreased the thermal conductivity in accordance with Ramamurthy [2]. Our observations agree with Ramamurthy’s; under lower cement content in the matrix the thermal conductivity is lower, with C70P30H0 exhibiting a remarkable thermal performance at 28 days. Overall, a large quantity of cement had a negative influence on thermal conductivity. Nonetheless, all the control samples had a density lower than 800 kg·m^−3^, a compressive strength higher than 2 MPa and a thermal conductivity lower than 0.2 Wm^−1^K^−1^, and, thus, achieved the specifications for the material.

### 3.4. Cost and CO_2_ Emissions Estimation

The most important factors for the applicability of any material are their economic and environmental impacts. Therefore, for the elaboration of these materials (HS-BFCs) the CO_2_ emissions and the cost were estimated on the base of each component’s quotation and technical supplier’s documents. To be able to evaluate the foam concretes produced, a rating of each foam concrete was achieved according to its performance based on 5 levels between the maximum and minimum values of each performance in this study (Table 4). Thermal behaviour is considered the most important factor here as air bubbles and hemp shives are incorporated into the cementitious matrix to improve the thermal strength of the concrete, and in decreasing order of importance is compressive strength, CO_2_ emissions and cost, to which we respectively associated a weight of 4, 3, 2 and 1 for a multi-criteria analysis (Table 5). CO_2_ emissions were calculated based on the average amount of CO_2_ of each raw material [47], and the cost taken into account was the one communicated by the supplier of the raw materials.

Table 5 presents the score and the classification of each formulation, giving a global evaluation of the concrete according to these four performances. The sum of multiplication of the weight of each performance with the rating presents the score of the performance advantage. For example, for the thermal conductivity presented in Table 5, C70P30H0 (Rating = 3) has a higher rating than C100P0H0 (Rating = 1), so C70P30H0 is the most insulating.

Cost per produced m^3^ of BFCs is high. More than 50% of the cost comes from high-dose additives (accelerator, superplasticiser and foaming agent). The optimisation of formulations would be necessary to reduce the addition of the admixture and then the cost.

Remarkably, the foamed concrete with 100% cement showed high thermal conductivity with a high CO_2_ emission value, while C70P30H0 exhibited low thermal insulation and high compressive strength together with moderate CO_2_ emissions. However, C70P30H0’s cost is high because of the metakaolin, and admixtures are expensive.

Among bio-based foamed concretes, C55P30H15 could also be considered as a good compromise as it had relatively low thermal conductivity (λ = 0.1285 Wm^−1^K^−1^), high compressive strength (Rc = 3.22 MPa) and approximately half the amount of CO_2_ emissions of C100P0H0. In conclusion, the effects of incorporating hemp shives in foamed concretes is mainly the decrease in thermal conductivity through the decrease in density. The resulting mechanical performance decrease was due to using the pozzolan additions to compensate for the HS incorporation.

## 4. Conclusions

This paper focused on non-autoclaved bio-based foamed concretes for semi-structural applications, evaluated in terms of specified parameters.

A high amount of cement increases thermal conductivity and CO_2_ emissions. Cement substitutions by pozzolanic additives and/or hemp shiv can improve insulating thermal performances.

The addition of hemp shives for a specific quantity may result in phase segregation. Moreover, if the BFC is cured at high relative humidity, hemp shiv slightly increases the density of the BFC over time by slowly absorbing water and retaining it in the material. Overall, hemp shiv decreases density, compressive strength and thermal conductivity.

Bio-based foam concretes (BFCs) with the same amount of hemp shives show a significant difference in compressive strength and thermal conductivity.

The bio-based foam concrete C55P30H15 is considered a bio-based lightweight concrete (ρ = 593 kg/m^3^) and a good thermal insulator (with a thermal conductivity of λ = 0.1285 Wm^−1^K^−1^), with a reasonably large compressive strength (Rc = 3.22 MPa) and CO_2_ emission values about half of the ones of a traditional foam concrete. Therefore, BFCs are potential alternative materials to target semi-structural concrete applications in the construction industries.

In addition, the right balance then has to be found between acceptable mechanical strengths conditioned mainly by cement contents, porosities for lightweight applications and porosity-HS amounts controlling thermal behaviour. Moreover, the pore structure, size and distribution of air bubbles will be important aspects to study, especially if one can help control the stability of air bubbles in the cementitious structure and then affect the mechanical and thermal conductivity.

## Figures and Tables

**Figure 1 materials-16-00591-f001:**
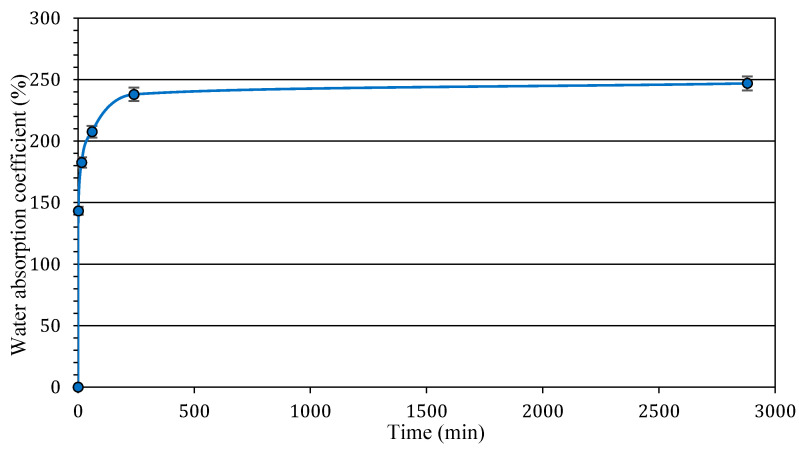
Water absorption kinetics of hemp shiv versus time.

**Figure 2 materials-16-00591-f002:**
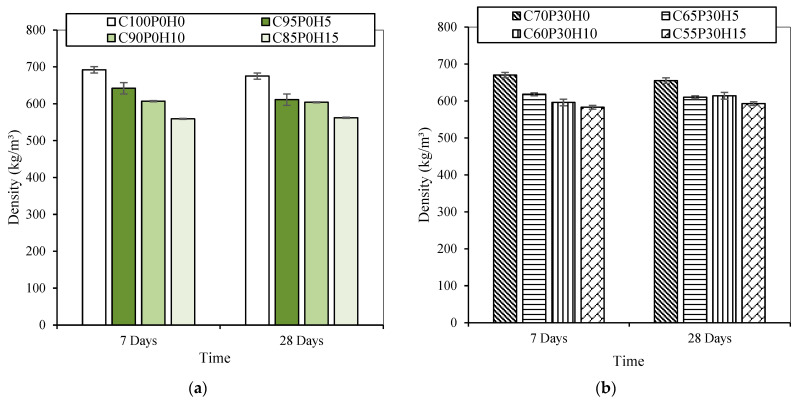
Density of all samples against time for (**a**) C100-BFCs and (**b**) C70-BFCs.

**Figure 3 materials-16-00591-f003:**
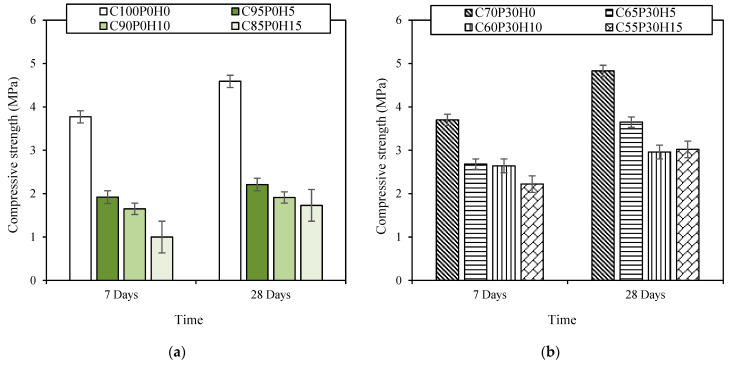
Compressive strength against time for (**a**) C100-BFCs and (**b**) C70-BFCs.

**Figure 4 materials-16-00591-f004:**
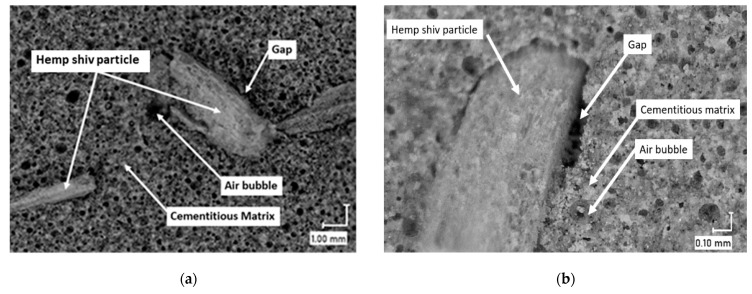
(**a**) Overview of the pore structure of the BFC material. (**b**) Interface between the cementitious matrix and the hemp shiv.

**Figure 5 materials-16-00591-f005:**
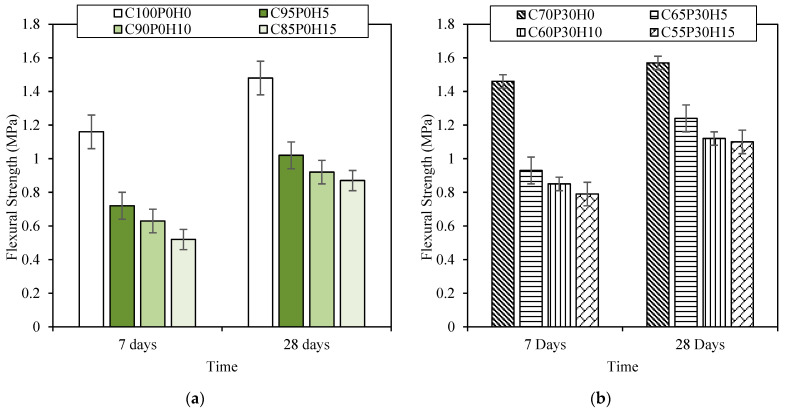
Flexural strength against time for (**a**) C100-BFCs and (**b**) C70-BFCs.

**Figure 6 materials-16-00591-f006:**
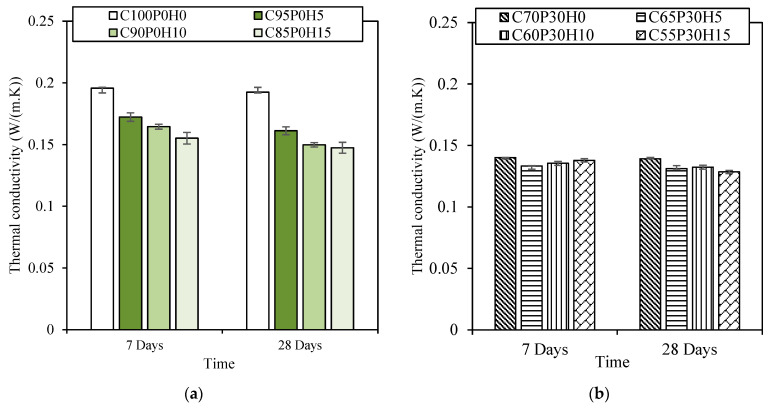
Thermal conductivity of all samples versus time for (**a**) C100-BFCs and (**b**) C70-BFCs.

**Table 1 materials-16-00591-t001:** Chemical composition of cement and pozzolans used in this study.

The Chemical Compound	Chemical Composition (%)		
	CEM I	GGBFS	MK
Calcium oxide (CaO)	64.17	38.6	0.2
Aluminium oxide (Al_2_O_3_)	4.44	12.3	24.1
Silicon dioxide (SiO_2_)	19.6	36.9	68.1
Ferric oxide (Fe_2_O_3_)	4	0.3	3.7
Sulphur trioxide (SO_3_)	2.6	2.1	-
Sodium oxide (Na_2_O)	0.07	-	0.1
Magnesium oxide (M_g_O)	1.25	7.5	0.2
Potassium oxide (K_2_O)	0.84	-	0.4
Others	3.03	2.3	3.2

**Table 2 materials-16-00591-t002:** Characteristics of admixtures used in the elaboration of BFC.

	Acc	SP	FAG
Consistency	Liquid	Liquid	Liquid
Colour	Yellow	Brown	Bright yellow
Density (g/cm^3^)	1.45 ± 0.01	1.05 ± 0.01	1.04 ± 0.02
Recommended dose *	1–1.5%	1–3%	-
Chlorides content	≤0.1%	≤0.1%	0.001%
pH	6 ± 1	6 ± 1	9
Solids Content (%)	61.5% ± 2.7%	30.5% ± 1.5%	30%

* Dose percentage is in respect to the total mass.

**Table 3 materials-16-00591-t003:** Foamed concretes composition and density.

Names of Mixes	Fresh Density (kg/m^3^)	Composition of Mixture (kg/m^3^)									Wt/B	Wb/B
		C	GGBFS	MK	HS	SP	Acc	FAG	Wh	Wt		
C100P0H0 *	891	700	-	-	-	14	7	2.1	0.00	168.00	0.24	0.24
C70P30H0	933	490	140	70	-	7	7	2.1	0.00	217.00	0.31	0.31
C95P0H5	858	665	-	-	2.2	14	7	2.1	5.57	165.17	0.25	0.24
C65P30H5	900	455	140	70	2.2	7	7	2.1	5.57	211.72	0.32	0.31
C90P0H10	825	630	-	-	4.5	14	7	2.1	11.14	162.34	0.26	0.24
C60P30H10	867	420	140	70	4.5	7	7	2.1	11.14	206.44	0.33	0.31
C85P0H15	792	595	-	-	6.7	14	7	2.1	16.70	159.50	0.27	0.24
C55P30H15	834	385	140	70	6.7	7	7	2.1	16.7	201.95	0.34	0.31

* C: 100% of cement, 0% of Pozzolanic additions and 0% of Hemp Shiv.

**Table 4 materials-16-00591-t004:** Rating of performances.

Rating	Thermal Conductivity (Wm^−1^K^−1^)	Compressive Strength (MPa)	Cost (€/m^3^)	CO_2_ Emissions (kg CO_2_/m^3^)
0	>0.2	0–1	>400	>600
1	0.175–0.2	1–2	300–400	500–600
2	0.15–0.175	2–3	200–300	400–500
3	0.125–0.15	3–4	100–200	300–400
4	0.1–0.125	>4	0–100	0–300

**Table 5 materials-16-00591-t005:** BFCs multi-criteria ranking.

Names of Mixes	Cost (€/m^3^)	CO_2_ Emissions (kg CO_2_/m^3^)	Ranking					
			Thermal Conductivity	Compressive Strength—28 days	CO_2_	Cost	Score ^b^	Global Rank ^c^
			Weight ^a^ = 4	3	2	1		
C100P0H0	259	665	1	4	0	2	18	7
C70P30H0	225	484	3	4	2	2	30	1
C95P0H5	249	632	2	2	0	2	16	8
C65P30H5	215	451	3	3	2	2	27	3
C90P0H10	239	599	3	1	1	2	19	5
C60P30H10	205	417	3	2	2	2	24	4
C85P0H15	229	565	3	1	1	2	19	5
C55P30H15	195	384	3	3	3	3	30	1

^a^ Weights are coefficients to give preference to the characteristic. ^b^ Score = sum (Rating × Weight). ^c^ Global Rank is the classification of BFCs from best (1) to worst (8).

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
