# Peer review of "Influence of Pozzolans and Hemp Shives on the Properties of Non-Autoclaved Foamed Concrete"

_materials, 2023, doi:10.3390/ma16020591_

Round 1

Reviewer 1 Report

In this paper, new foam concretes incorporating hemp shives without use of autoclaving have been investigated. I recommend the publication of this manuscript after minor revision.

1) Line 90 the superscript of “-1”

2) It’s better to provide the model numbers and producers of Acc, SP and FAG in section 2.1.

3) How about the shape and size of samples for the test of thermal conductivity?

4) Please introduce the physical meaning of Rf/Rc in Line 229-233.

5) Line 263 “in a foam concrete” should be revised as “in foam concrete”

6) Line 287 “table” should be revised as “Table”.

7) The format of references should be unified (such as the author’s name in Ref. [26] and [27]).

Author Response

1) Line 90 the superscript of “-1”

- Correction made (L. 91).

2) It’s better to provide the model numbers and producers of Acc, SP and FAG in section 2.1.   

- Details added (L. 100 – 107).

3) How about the shape and size of samples for the test of thermal conductivity?

- Details added (L. 151 – 152).

4) Please introduce the physical meaning of Rf/Rc in Line 229-233.

- Introduction added (L. 229 – 233).

5) Line 263 “in a foam concrete” should be revised as “in foam concrete”

- Correction made (L. 268).

6) Line 287 “table” should be revised as “Table”.

- Correction made (L. 292).

7) The format of references should be unified (such as the author’s name in Ref. [26] and [27]).

- Correction made (L. 395 and L. 397).

Reviewer 2 Report

In this paper, the authors investigated the influence of pozzolanic and hemp on the properties of foam concrete. The results were interesting and the mechanism explanation was reasonable. But there were still some problems should be revised.

(1) The title is too general and does not summarize the article well. Recommended modification.

(2) In table 2, why not keep the same significant numbers?Such as 1.45 and 1.055.

(3) The authors described that “ pozzolan additions might help a better pores filling giving rise to slightly larger thermal conductivities”. The authors should provide data of porosity to verify the mechanism.

(4) The test condition of thermal conductivities should be given, especially the test temperature.

(5) Many factors could affect thermal conductivity. Whether the pore size is one of the influencing factors in this experiment?

(6) The y-axis font in Figure 2 is inconsistent with the other figures, please revise it. There are many formatting inconsistencies in the figures, the authors should check carefully.

Author Response

(1) The title is too general and does not summarize the article well. Recommended modification.

- The title has been changed by “Influence of pozzolans and hemp shive on the properties of non-autoclaved foam concrete”.

(2) In table 2, why not keep the same significant numbers ? Such as 1.45 and 1.055.

- Correction made.

(3) The authors described that “ pozzolan additions might help a better pores filling giving rise to slightly larger thermal conductivities”. The authors should provide data of porosity to verify the mechanism.

- Reference added (L. 265).

(4) The test condition of thermal conductivities should be given, especially the test temperature.

- Details added (L. 164 – 166).

(5) Many factors could affect thermal conductivity. Whether the pore size is one of the influencing factors in this experiment?

- Precisions have been added in the conclusion (L. 336 – 339).

(6) The y-axis font in Figure 2 is inconsistent with the other figures, please revise it. There are many formatting inconsistencies in the figures, the authors should check carefully.

- Figures are revised.
